# Formation Optimization, Characterization and Antioxidant Activity of *Auricularia* *auricula-judae* Polysaccharide Nanoparticles Obtained via Antisolvent Precipitation

**DOI:** 10.3390/molecules27207037

**Published:** 2022-10-18

**Authors:** Yemei Dai, Yuan Ma, Xiaocui Liu, Ruyun Gao, Hongmei Min, Siyu Zhang, Siyu Hu

**Affiliations:** Sichuan Key Laboratory of Food Biotechnology, School of Food and Bioengineering, Xihua University, Chengdu 610039, China

**Keywords:** *Auricularia auricula-judae* polysaccharide, nanoparticles, antisolvent precipitation, optimization, characterization, antioxidant activity

## Abstract

*Auricularia auricula-judae* polysaccharide (AAP)-based nanoparticles (NPs) prepared via an anti-solvent precipitation approach were studied. Response surface methodology (RSM) design was carried out on the basis of single factor experiments, using average size and polydispersity index (PDI) as indicators. The optimal preparation conditions were determined to include an AAP concentration of 1 mg/mL, a pH of 8, and an anti-solvent/solvent volume ratio of 6. The average particle sizes of the AAP-NPs, PDI and electrical characteristic (ζ-potential) were found to be 150.27 ± 3.21 nm, 0.135 ± 0.012 and −31.10 ± 0.52 mV, respectively. Furthermore, Fourier transform infrared spectroscopy (FTIR) was used to determine the chemical structure of the AAP-NPs. It was observed that the intensity of AAP-NPs in the wide spectral band of 3000–3750 cm^−1^ was significantly stronger than that of the AAP, as was the characteristic peak of carboxyl anion, and the characteristic band moved to shorter wavelengths. Subsequent thermogravimetric analysis showed that the antisolvent precipitation method improved the thermal stability of the AAP, while scanning electron microscopy (SEM) and X-ray diffraction (XRD) showed that the morphology of AAP-NPs was uniform and well-distributed, and that their single crystal structures had remained unaffected during the process. Moreover, the DPPH and ABTS scavenging activities of AAP-NPs were increased, and the IC50 values were 0.544 ± 0.241 mg/mL and 0.755 ± 0.226 mg/mL, respectively.

## 1. Introduction

*Auricularia auricula-judae* (*A. auricula*), of the genus *Auricularia*, family *Auriculariaceae*, order *Auriculariales*, class *Agaricomycetes*, and phylum *Basidiomycota* [1], is a common edible fungus with rich nutritional value found mainly in the Northern Hemisphere. Modern medical research [2,3,4,5] has shown that *A. auricula* can also provide extensive medicinal benefits. Its main active component, *A. auricula* polysaccharide (AAP), has been found to exhibit numerous physiological functions, including antioxidation [6,7], and antibacterial [8], hypoglycemic [9] and anti-cardiovascular activities [10]. AAP shows little toxicity or side effects in vivo, indicating that it has application value in the development of food and health products [11]. However, as a long-chain carbohydrate formed by the complex polymerization of various neutral sugars or ursolic acids through glycosidic bonds [12], polysaccharides exhibit poor pharmacokinetic behavior, resulting in low drug delivery and bioavailability in vivo [13].

The conversion of bioactive polysaccharides into nanoparticles (NPs) is an effective way to improve their comprehensive utilization. NPs exhibit promising advantages such as good cell penetration, targeting ability and biocompatibility and, consequently, are increasingly the subject of scientific investigation [14]. Due to their small particle size, large specific surface area and good biocompatibility, nanoparticles have been widely used in biomedical food industries, in applications such as nanocarriers and edible coatings. Generally speaking, the main methods used in the preparation of NPs include ion cross-linking [15,16], anti-solvent precipitation [17], emulsion cross-linking [18,19], chemical cross-linking [20,21], condensation, and spray drying [22]. Anti-solvent precipitation involves the addition of a solution of a drug or polymer (usually in a non-volatile solvent) to its antisolvent which leads to the precipitation of amorphous solid dispersions, provided the drug or polymer is insoluble in the antisolvent [23]. While there have been reports on the preparation of AAP-NPs using the ion cross-linking method, such as the studies by Xiong et al. [24,25] in which a new type of polyelectrolyte complex NPs were prepared using negatively charged *A. auricula* polysaccharides and positively charged chitosan as materials, to the best of our knowledge little information has thus far been made available on the preparation and biological activities of *A. auricula* polysaccharides nanoparticles (AAP-NPs). Compared with other methods in the preparation process, including the ion cross-linking method, the antisolvent precipitation method requires no special equipment, is low in cost, requires fewer types of reagents, and does not easily pollute samples, which is why it has recently attracted widespread attention [26]. Kakran et al. [27] prepared quercetin nanoparticles via injection using the anti-solvent precipitation method, and results showed that the dissolution rate and solubility of quercetin obtained using this method were improved in simulated intestinal fluid (SIF, pH 6.8) compared with the pure drugs. Chen et al. [28] developed oxidized lotus root starch (OLRS) nanoparticles in the hope of improving the stability and antioxidant activity of luteolin through this method. The results showed that OLRS NPs stabilized luteolin in simulated gastric juice and continuously released luteolin in simulated intestinal juice, and the antioxidant capacity was improved. Charoenchaitrakool et al. [29] used acetone as an organic solvent to dissolve the copolymers of mefenamic acid (MEF), paracetamol (PAR) and nicotinamide (NIC), with carbon dioxide as an anti-solvent. The new eutectic of MEF-PAR-NIC was successfully prepared and, in comparison with unprocessed MEF, the dissolution rate of MEF was increased 23.9 times and 20.8 times, respectively, under two different experimental conditions. The results of these studies suggest that the preparation of nanoparticles via the antisolvent precipitation method could result in different characteristics from the corresponding bulk materials while increasing the specific surface area of the material. However, this approach has not yet been applied to fungal polysaccharides.

Due to its simple operation and low cost, anti-solvent precipitation has more industrial application value compared with other nanoparticle preparation methods. Since this method had not hitherto been applied to mushroom polysaccharides, response surface methodology was used to optimize the preparation conditions of AAP-NPs, whereafter the particle size, PDI and ζ-potential of the prepared AAP-NPs were determined and the optimal preparation process of AAP-NPs via anti-solvent precipitation was obtained. Furthermore, scanning electron microscopy (SEM), Fourier transform infrared spectroscopy (FTIR), X-ray diffraction (XRD), thermogravimetric analysis and in vitro antioxidant activity analysis were performed for comparative analyses with the original AAP. This study enriches the existing preparation methods of mushroom nanoparticles and has reference significance for the industrial preparation of mushroom nanoparticles.

## 2. Results and Discussion

### 2.1. Selection of Factors and Their Levels via Single Factor Analysis

Prior to the optimization of the preparation parameters, the single factor effects of AAP-NPs average particle size as well as the PDI between AAP concentration, pH value and (*v*/*v*) were studied.

As shown in Figure 1a, within the AAP concentration range of 0.5~2.5%, the average particle size and PDI value of the AAP-NPs decreased initially and then increased. The antisolvent precipitation method produces nanoparticles in a ‘bottom-up’ self-assembly process. The precipitation process includes (i) supersaturation generation, (ii) nucleation and (iii) subsequent nuclei growth [30,31]. Higher drug concentrations can lead to higher supersaturation, resulting in faster nucleation rates [32]. In this study, when the AAP concentration was 1%, the average particle size of the AAP-NPs reached a minimum of 161.102 ± 4.932 nm, the molecular weight distribution of the AAP-NPs was the most uniform, and the PDI was also lowest at 0.212 ± 0.013. However, high supersaturation can also accelerate particle collisions, producing particles with a larger diameter. At higher drug concentrations, a large number of nuclei are formed at the interface of the two phases, and particle agglomeration is better than nucleation since it results in a wider molecular weight distribution of particles [33]. Thus, 1% was found to be the optimal concentration of the AAP-NPs prepared via the antisolvent precipitation approach selected by single factor experiment.

The average particle size and PDI of AAP-NPs in the range of pH = 5.0~9.0 are shown in Figure 1b. The size of AAP-NPs was found to be stable at 150~200 nm under alkaline and neutral conditions. When the pH reached 7, the particle size and PDI of the AAP-NPs reached their minimum values, at 165.391 ± 5.519 nm and 0.221 ± 0.009, respectively. However, particle size increased significantly under acidic conditions. These experimental results are consistent with those reported by Qin et al. [34], and indicate that the stability of AAP-NPs under neutral and alkaline conditions was better than that under acidic conditions. The reason for this phenomenon may be that the glycosidic bonds in the polysaccharides only ensured their stability in alkaline and neutral environments. However, the increase of the concentration of hydronium ion in an acidic medium has been shown to accelerate the hydrolysis and degradation process of glycosidic bonds, subsequently leading to the shortening of the cellulose chain of *A. auricula* polysaccharides, manifested as the decrease in the degree of AAP-NP polymerization [35].

The antisolvent: solvent volume ratio is a key index in the preparation of nanoparticles via antisolvent precipitation [36]. As shown in Figure 1c, with the increase in the antisolvent: solvent ratio (*v*/*v*), the AAP-NPs first decreased slowly and then increased sharply. When the antisolvent: solvent ratio (*v*/*v*) was 6:1, the mean particle size and PDI of AAP-NPs reached their minimum values or 180.591 ± 1.173 nm and 0.195 ± 0.015, respectively. Previous studies have shown that the solubility of polysaccharides in ethanol gradually decreases as the degree of polymerization increases [37]. Combined with the principle of preparing AAP-NPs via antisolvent precipitation, under the condition of various antisolvent: solvent volume ratios, the components of the AAP precipitated by local saturation may differ. At a lower antisolvent: solvent volume ratio, for example, an AAP with a low degree of polymerization is precipitated and so the particle diameter is smaller, whereas under the condition of a higher volume ratio, an AAP with a high degree of polymerization is mainly precipitated.

Through the single factor experimental design herein, the concentration of 1% AAP, pH of 7 and antisolvent: solvent ratio (*v*/*v*) of 6:1 were ultimately selected as the central parameters for the BBD, and the average particle size and PDI were taken as the response values to further optimize the preparation of AAP-NPs.

### 2.2. RSM Optimization of Operating Parameters

#### 2.2.1. Model Building and ANOVA

Compared with the classical central composite design (CCD), BBD requires less experimental data in the case of three factors [38] and, consequently, was selected for application in this study. A three-factor and three-level Box-Behnken response surface test was designed using Design-Expert V. 8.1.5 (Stat-Ease Inc., Minneapolis, USA), and the (A) AAP concentration, (B) pH and (C) antisolvent: solvent volume ratio were optimized. The experimental parameters, average size and PDI of the AAP-NPs are shown in Table 1, with the ANOVA results presented in Table 2.

Based on the results of the multiple regression analysis, the quadratic regression equation of the AAP-NPs’ (Y_1_) average size and (Y_2_) PDI with (A) AAP concentration, (B) pH and (C) antisolvent: solvent volume ratio were obtained as follows:Y_1_ = 153.07 − 2.51A − 2.03B − 1.25C − 3.32AB + 1.52AC + 4.44BC + 15.75A^2^ + 8.27B^2^ + 10.24C^2^(1)
Y_2_ = 0.14 − 0.023A − 0.034B − 2.273E − 0.003C − 0.018AB − 0.022AC + 0.016BC + 0.11A^2^ − 0.069B^2^−0.049C^2^(2)

As shown in Table 2, the Model F-values of (Y_1_) average size and (Y_2_) PDI were 99.86 and 58.63, respectively, and both had *p*-values < 0.0001, indicating that both of the two models had reached the highly significant level and had statistical significance. In addition, the ‘lack of fit’ of both Model 1 and Model 2 had *p*-values > 0.05, with no significant influence, demonstrating that there were no misfitting factors and that the test sites could be described by the model. Furthermore, the regression equation could well explain the results and predict the best preparation conditions. The correction coefficients (R^2^) were, successively, 0.9923 and 0.9869, both of which results were above 95%. Thus, the dependent variable was shown to have a significant impact on the independent variables (*p* < 0.01). The adjusted determination coefficient (*R_adj_*^2^) of Model 1 was 0.9823, and the coefficient of variation (C.V.%) was 0.87, indicating that only 0.87% of the total variation could not be explained by the model and that it, thus, had achieved a high degree of fit. Moreover, the ratio of adeq. precision of Model 1 was 27.915 more than 4, demonstrating that this Mode l could be used to navigate the design space. The same results and conclusion were obtained with Model 2.

The significance of each factor (A-AAP concentration, B-pH and C-antisolvent: solvent volume ratio) was subsequently tested. In Model 1, the one degree terms A and B were found to have significant effects on the average size (Y_1_) of the AAP-NPs (*p* < 0.01), the interaction terms AB and BC also had significant effects on Y_1_ (*p* < 0.01), while the second-order terms A^2^, B^2^ and C^2^ had significant effects on the average size (*p* < 0.01). In Model 2, the one degree terms A, B and C had extremely significant effects on the PDI (Y_2_) of the AAP-NPs (*p* < 0.01), and the *F*-values revealed that the contributions of the factors in this experiment were A > B > C. The interaction terms AB and AC had significant effects on PDI (*p* < 0.05), and the second-order terms were extremely significant (*p* < 0.01).

#### 2.2.2. Three-Dimensional (3D) Response Surface Diagram Analysis

The 3D response surface plot provides an intuitive interpretation of the interaction between two independent variables. The degree of influence of the two variables on the response value was determined by observing the inclination degree of the surface. The higher the inclination degree of the 3D response surface graph, the steeper the slope and the more significant the interaction [39,40].

As can be seen in the 3D response surface diagram of the models (Figure 2), when the values of the three factors, (A) AAP concentration, (B) pH and (C) antisolvent: solvent volume ratio, were small, the response surface was steep, indicating that the influence of each factor on the (Y_1_) average size and (Y_2_) PDI of the AAP-NPs was obvious. However, when the value was large, the opposite occurred. Moreover, intense interactions between factors A and B, and B and C of Model 1 (Figure 2a,c), as well as between A and B, and A and C of Model 2 (Figure 2d,e) were observed visually, all of which were consistent with the results of the ANOVA.

#### 2.2.3. Prediction of Optimal Response Surface

The minimum values of average size and PDI were applied to solve the regression equation using Design-Expert V. 8.0.6, thereby obtaining the optimal preparation conditions based on the antisolvent precipitation approach for AAP-NPs. The optimal combination of factors predicted by the software were an AAP concentration of 1.05 mg/mL, a pH of 8.01, and an antisolvent: solvent volume ratio of 6.01:1. The predicted average size and PDI were 152.95 nm and 0.141, respectively. The optimized experimental conditions were refined for real application as follows: an AAP concentration of 1 mg/mL, pH of 8, and antisolvent: solvent volume ratio of 6:1.

### 2.3. Morphology and Size Analysis

Particle size and dispersion are two key factors affecting the therapeutic potential of nanocarriers [41]. The best applicable range of PDI for the algorithm was found to be between 0.08 and 0.7, indicating that the sample to be tested was a moderately dispersed system. The larger the absolute value of ξ-potential, the stronger the repulsive force between particles and, consequently, the more stable the dispersion of particles in the system [42]. It is generally believed that polysaccharide solutions are relatively stable when the absolute value of ξ-potential is approximately 30 mV. The particle size distributions of AAP-NPs in triplicate experiments are shown in Figure 3. In this study, the average size of AAP-NPs was 150.27 ± 3.21 nm, and the PDI was 0.135 ± 0.012, which were in good agreement with the response surface prediction results. In addition, the ζ-potential of the AAP-NPs was −31.10 ± 0.52 mV. Zhang et al. [43] used the same method to prepare *Sargassum pallidum* polysaccharide nanoparticles and obtained an average particle size, PDI and ξ-potential of 229.63 nm, 0.407 and −28.43 mV, respectively. The experimental results showed that the AAP-NPs prepared herein via response surface had a small particle size and good dispersion stability, indicating that the response surface had played an effective role in optimizing their preparation conditions.

In addition, the apparent morphologies of the AAP-NPs and AAP as observed via SEM is shown in Figure 4, in which it can be seen that the AAP without antisolvent precipitation was irregular sheet or rod shaped, with a large overall volume, while the rod-like AAP-NPs with uniform morphology and good distribution had been successfully synthesized. In conclusion, antisolvent precipitation effectively reduced the volume and increased the specific surface area of the AAP. Combined with the preparation process and SEM image analysis, the formation process of AAP-NPs can be described as follows: First, the AAP solution was diluted in a volume of absolute ethanol in which, due to supersaturation, it precipitates into a flocculent shape; subsequently, under the action of constant speed magnetic stirring, the flocculating polysaccharide is constantly wound and contracted, forming tiny nanoparticles.

### 2.4. FTIR Spectra Analysis

The effect of the antisolvent precipitation on the structure of AAP, as measured via FTIR, is shown in Figure 5. In general, the spectral features of the AAP-NPs and AAP were found to be similar, with the exception of some characteristic bands that showed changes in absorbance and wave number. The AAP-NPs and AAP shared a broad band in the 3000–3750 cm^−1^ region, and both had a strong characteristic absorption peak at 3397 cm^−1^, which was caused by the O-H stretching vibration of the polysaccharide hydroxyl group. In addition, a weak absorption peak at 2925 cm^−1^ was caused by the C-H stretching vibration of the polysaccharide alkyl group. The typical groups of polysaccharides are hydroxyl and alkyl [44,45]. The above results indicated the consistency of the AAP and AAP-NPs in the characteristic functional groups. Both the asymmetric (1700~1600 cm^−1^) and symmetric (1400~1300 cm^−1^) stretching of carboxyl anions exhibited characteristic peaks, indicating the presence of carboxyl groups and uronic acids in the samples, which were also responsible for the antioxidant properties of the AAP and AAP-NPs [46]. Intuitively, it was evident that the strength of AAP-NPs in the broad spectral band of 3000~3750 cm^−1^ was significantly greater than that of the AAP. Zhang et al. [47] speculated that the hydrogen bonds between polysaccharide molecular chains become stronger during nanotization. Interestingly, the intensity of the characteristic peak of the carboxyl anion of AAP-NPs in this study was significantly stronger than that of the AAP, and the characteristic band shifted to shorter wavelengths, indicating that the carboxyl anion in the polysaccharide molecular chain had been strengthened in the process of glycogenation. Therefore, it was speculated that AAP-NPs could have higher antioxidant activity than AAP, which was subsequently verified in the in vitro free radical scavenging activity experiment.

### 2.5. XRD Analysis

XRD phase analysis is a technique that uses the diffraction effect of X-ray in crystalline materials to analyze the structure of materials as either crystal or amorphous [48]. As shown in Figure 6, the multiple independent diffraction patterns of AAP-NPs and AAP showed no distinct differences, indicating that their crystal structures were the same and that the crystal structure of the AAP-NPs was not influenced by the antisolvent precipitation nanotization. Zhang et al. [43] used absolute ethanol as the antisolvent to prepare *Sargasso pallidum* polysaccharide NPs, while Park et al. [49] used distilled water as the antisolvent to prepare carbamazepine NPs, and the crystallinity of neither of these NPs was found to be affected by the process. In addition, in this study, the XRD pattern of both the AAP-NPs and AAP exhibited a sharp and narrow peak (2θ = 28.67°), which indicated that both had stable crystal structures. This result is consistent with that reported by Ren et al. [50], in which the XRD of quinoa (*Chenopodium quinoa* Willd.) polysaccharide QPI-Ⅰ had an obvious diffraction peak at 28.40°, which inferred that the QPI-Ⅰ had a stable single crystal structure.

### 2.6. Thermostability Analysis

Figure 7a shows that the thermogravimetric (TG) curves of the AAP-NPs and AAP within the experimental temperature range were similar, and could be summarized into three approximate stages. During the first stage, the mass losses of the AAP-NPs and AAP were approximately 9.47% and 10.24%, respectively, when the temperatures of the samples rose from 25 °C to 150 °C as a result of the evaporation of free and bound water in the polysaccharides [51]. In the second stage, the AAP-NPs and AAP exhibited obvious mass losses, mainly in the temperature range of 200 °C to 400 °C. The mass loss rates of the AAP-NPs and AAP were approximately 53.43% and 66.02%, respectively, and may have been caused by the glycan degradation of polysaccharides in this stage [52,53]. The third stage of thermal degradation was from 450 °C to 550 °C, during which both the AAP-NPs and AAP were calcined into ash. The final residual amounts at 600 °C were 10.5% and 5.63%, respectively. Corresponding to the TG curve, a DTG curve reflects the variation in the sample mass loss rate with temperature, and the peak value represents the temperature at which a sample reaches its maximum mass loss rate. In the DTG curves herein (Figure 7b), a peak in the AAP curve was evident at 480.01 °C, however, the peak shifted to 485.7 °C corresponding to the AAP-NPs curve, indicating that the antisolvent precipitation method had improved the thermal stability of the AAP. This may have been due to the strong electrostatic interaction between the hydrogen bonds in the AAP-NPs particles. Thermostability analysis laid a foundation for the application of AAP-NPs in thermal processing such as sterilization and baking [52].

### 2.7. Free Radical Scavenging Activity Assays

Polysaccharides isolated from natural sources have been shown to have potent antioxidant capabilities in vitro [54]. DPPH is a relatively stable free radical, and exogenous antioxidants can pair with free electrons to form stable presence groups, thereby weakening absorption. Antioxidant activity is evaluated indirectly by measuring the degree of absorption reduction. Due to its high stability, experimental feasibility and low cost, the DPPH method is often used in experiments to evaluate the antioxidant activity of chemical substances [55]. The DPPH radical scavenging activity of the AAP-NPs in this study compared with that of the VC is shown in Figure 8a. The AAP-NPs, AAP and VC all showed good linear dose-response relationships within the experimental concentration range. At each concentration point, the AAP-NPs showed higher DPPH free radical scavenging rates than those of the AAP, but both were consistently lower than those of the positive control VC. Under the conditions of low concentration (0.1 mg/mL) and high concentration (2 mg/mL), the DPPH free radical scavenging rate of the AAP-NPs was higher than that of the AAP (33.333 ± 3.162%) and (24.860 ± 2.051%), respectively. The half maximal inhibitory concentrations (IC_50_) of the AAP-NPs, AAP and VC were 0.544 ± 0.241 mg/mL, 1.004 ± 0.236 mg/mL and 0.060 ± 0.207 mg/mL, respectively. These results showed that the DPPH scavenging activity of A. auricula polysaccharides could be improved by antisolvent precipitation. Chen et al. [56] reported a similar effect when they prepared Mori Fructus nanoparticles using the same method, with their results indicating a significant improvement in the antioxidant activities of the nanoparticles.

The in vitro antioxidant activities of the AAP-NPs were subsequently further determined via the measurement of ABTS+ radical scavenging activity. Mixed with ABTS+ radicals, samples with antioxidant properties will generate colored substances with free radicals, which subsequently fade the total system and reduce its light absorption value, thereby reflecting the strength of the antioxidant ability [57]. As shown in Figure 8b, the AAP-NPs, AAP and VC in this study all exhibited dose-dependent quenching of ABTS+ radicals in the experimental concentration range. Based on IC_50_, the rank order of ABTS+ radical scavenging rates were as follows: AAP (IC_50_ mg/mL = 1.059 ± 0.226) > AAP-NPs (IC_50_ mg/mL = 0.755 ± 0.226) > VC (IC_50_ mg/mL = 0.063 ± 0.327). These experimental results, therefore, prove that the antisolvent precipitation treatment similarly increased the ABTS+ radical scavenging rate of the agaric polysaccharides.

## 3. Materials and Methods

### 3.1. Chemicals and Reagents

*A. auricula* polysaccharides were purchased from Shanxi Ciyuan Biotech Co., Ltd. (Shaanxi, China). Potassium bromide (SP, 99%), DPPH (SP, 98%), ABTS diammonium salt (SP, 98%), oxalic acid (98%), sodium hydroxide, ascorbic acid (VC) (analytical standard product, HPLC ≥ 99%) were purchased from Yuanye Biotechnology Co., Ltd. (Shanghai, China). N-butanol, ethyl alcohol, potassium persulfate and other analytic-grade reagents were obtained from the Kelong Chemical Reagent Factory (Chengdu, China).

### 3.2. Formation of AAP-NPs

The *A. auricula* AAP-NPs were prepared via antisolvent precipitation, according to the experimental methods reported by Qin et al. [34] and Machmudah et al. [58], with slight modifications. Based on the differences in solubility between polysaccharides in deionized water and those in absolute ethanol, deionized water was used as the solvent while absolute ethanol was used as the antisolvent. Briefly, the *A. auricula* polysaccharides were mixed with deionized water and the solution was fully dissolved by magnetic stirring at room temperature (25 °C) for 1 h. The solution was then filtered through a 0.22 μm microporous filter membrane, and the pH value of polysaccharide solution was subsequently adjusted using sodium hydroxide or oxalic acid. Under constant speed mechanical stirring for 2 h with a magnetic stirrer, 2 mL polysaccharide solution was added to a certain volume of absolute ethanol. In this process, AAP was constantly supersaturated and precipitated into filaments, and then wound and contracted under the action of magnetic stirring at a constant speed until, finally, AAP-NPs were formed and distributed uniformly in the solution. The suspension was then centrifuged, the precipitate was rinsed with absolute ethanol to remove excess water, and the AAP-NPs were obtained via lyophilization. The formation process of AAP-NPs is shown in Figure 9.

### 3.3. Determination of Average Particle Size, Polydispersity Index and Electrical Characteristics of AAP-NPs

The average particle size, PDI and ζ-potential of the AAP-NPs were determined via the dynamic light scattering (DLS) technique using a zetasizer (Nano ZEN3600, Malvern Instruments, Worcestershire, UK). Prior to measurement, to avoid multiple scattering effects, the nanoparticle dispersion was diluted with ultra-pure water (0.1~0.5%) and equilibrated in a measuring chamber at 25 ± 1 °C. All measurements were performed in triplicate.

### 3.4. Optimization of AAP-NPs Formation

#### 3.4.1. Single Factor Designs

The general preparation of AAP-NPs via the antisolvent precipitation approach was performed as described above. The concentration and pH of the polysaccharide solution, along with the solvent: antisolvent ratio, were the three main factors affecting the formation of the nanoparticles [59]. The average particle size and PDI of the AAP-NPs measured via the zetasizer were used as evaluation indexes to investigate the effects of the *A. auricula* polysaccharide solution concentration, pH value and antisolvent: solvent ratio on the average particle size formation and dispersion of the AAP-NPs. Each experiment was performed in triplicate.

When investigating the impact of the AAP solution concentration in a range of 0.5~3%, the pH value of the AAP solution and antisolvent: solvent ratio were set to 7 and 6:1, respectively.

When investigating the impact of pH value of the AAP solution in a range of 5~9, the concentration of the AAP solution was set to 2%, and the antisolvent: solvent ratio was kept at 6:1.

When investigating the impact of the antisolvent: solvent volume ratio in a range of 4:1~8:1, the concentration and pH value of AAP solution were set to 2% and 7.4, respectively.

#### 3.4.2. Box–Behnken Design (BBD)

The relevant factors affecting the formation of AAP-NPs were preliminarily determined on the basis of the single factor experimental results. RSM was subsequently applied to optimize the formation process using Design-Expert.V8.0.6.1 (Stat-Ease Inc., Minneapolis, MN, USA). According to the Box-Behnken central combination design principle, a three-level-three-factor BBD was employed to evaluate the combined effect of the three independent variables affecting the average particle size and PDI of AAP-NPs, namely (A) the AAP concentration, (B) pH value and (C) antisolvent: solvent ratio. The test factors and levels are presented in Table 3.

### 3.5. Scanning Electron Microscopy

The morphology of the AAP-NPs was analyzed using a Hitachi 7700 SEM (Tokyo, Japan), with AAP as the control. The samples were fixed on copper plates and sprayed with gold. The microscopic morphology of the samples was observed and recorded using scanning electron microscopy under high vacuum conditions with an accelerating voltage of 15 kV and image magnification of 2000 to 50,000.

### 3.6. Fourier Transform Infrared Spectroscopy

The chemical structures of the AAP-NPs and the AAP were measured via FTIR (Tensor 27, Jasco Inc., Easton, MD, USA). First, the AAP-NPs and AAP were ground with agate mortar, to which dried chromatography-pure KBr powder was added. The mixture was then placed evenly into the mold and pressed into pieces on the tablet press, after which infrared spectrum analysis was carried out with FTIR. The KBr powder was used as the background, and the spectral domain was set to be between 4000 and 500 cm^−1^.

### 3.7. X-ray Diffraction

The crystal morphologies of the AAP and AAP-NPs were determined using an XRD analyzer (Empyrean, Malvern Panalytical, Shanghai, China). Voltage and current of the copper target were set as 40 kV and 40 mA, respectively. Diffraction temperature was kept at 25 °C and the samples were collected in the 2θ range from 5–40°.

### 3.8. Thermostability

Thermogravimetric analyzer (TGA-50, Shimadzu, Kyoto, Japan) was used to study the thermodynamic properties of the AAP-NPs, with AAP as the control. Using accurately weighed 3~5 mg samples. The experimental temperature range was 25~550 °C, the heating rate was 10 °C/min, and the nitrogen flow rate was 50 mL/min.

### 3.9. DPPH Radical Scavenging Activity

The DPPH activity was measured according to a previously reported method, with some modifications [60]. Initially, a series of concentration gradient test sample solutions (1.0 mL, 0.1~2.0 mg/mL) were mixed with 300 μL DPPH solution (0.4 mmol/L in EtOH) and 2.0 mL distilled water. The reaction mixture was shaken well and then left in the dark for 30 min at room temperature. The resulting solution was measured at 517 nm, and AAP and VC (0.05~0.30 mg/mL) were used as controls. The DPPH radical scavenging rate was calculated using Equation (3):DPPH scavenging rate (%) = [A_2_ − (A_1_ − A_3_)]/A_2_ × 100%(3)
where A_1_ is the absorbance of the sample (DPPH solution with sample or controls), A_2_ is the absorbance of the control (DPPH solution with deionized water), and A_3_ is the absorbance of the blank (deionized water with sample or controls).

### 3.10. ABTS Radical Scavenging Activity

The ABTS radical scavenging rate was determined using a previously described method, with minor modifications [61]. In brief, 20 mL ABTS (7 mmol/L) solution and 352 μL (140 mmol/L) potassium persulfate solution were mixed by vortex and then left to stand in the dark for 16 h at room temperature (25 ± 2 °C). Subsequently, 4 mL ABTS stock solution was drawn and the absorbance at 734 nm was adjusted to 0.700 ± 0.02 with distilled water. Thereafter, various concentrations (200 μL, 0.1~2.0 mg/mL) of AAP-NPs solutions were mixed with 3 mL of a diluted ABTS solution in the dark for 30 min at room temperature. The absorbance was measured at 734 nm, as were the controls. The ABTS radical scavenging rate was calculated using Equation (4):ABTS scavenging rate (%) = [A_2_ − (A_1_ − A_3_)]/A_2_ × 100%(4)
where A_1_ represents the absorbance of the sample (diluted ABTS solution with sample or controls), A_2_ is the absorbance of the control (diluted ABTS solution with deionized water), and A_3_ is the absorbance of the blank (deionized water with sample or controls).

### 3.11. Statistical Analysis

In addition to the RSM, all statistical analyses were conducted using SPSS software (IBM SPSS 24, SPSS Inc., NY, USA). Data were analyzed via one-way analysis of variance (ANOVA) followed by Duncan’s multiple range test and least-significant difference (LSD) to determine significant differences among the means at an α = 0.05 level. The results were expressed as mean ± standard deviation. The graphics were created using Origin Pro software (93E version, OriginLab Corporation, MA, USA).

## 4. Conclusions

In this study, *A. auricula* polysaccharide nanoparticles were successfully prepared via the anti-solvent precipitation method, and the preparation conditions were optimized. The structure of the AAP-NPs prepared in this paper was that of a single crystal, which was not affected by the process of anti-solvent precipitation. Furthermore, the speculation that the process of antisolvent precipitation would enhance the hydrogen bonds between the molecular chains of AAP and strengthen the carboxyl anions was verified via subsequent thermogravimetric analysis and in vitro antioxidant tests. These results, thus, prove the suitability of AAP-NPs for application as antioxidants in the food- and health-related industries.

## Figures and Tables

**Figure 1 molecules-27-07037-f001:**
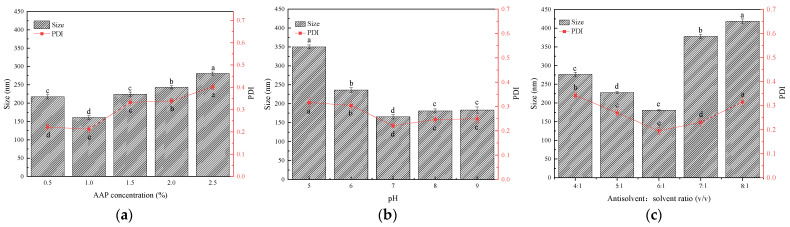
Effects of (**a**) different concentration of AAP, (**b**) solution pH levels and (**c**) antisolvent: solvent (*v*/*v*) ratios on the mean average particle size and PDI of AAP-NPs. Three independent experiments were carried out for each analysis. The data were measured by zetasizer, and results were expressed as a mean ± SD (*n* = 3). Values with different letters are those found to be significantly different in the Duncan’s multiple range test (*p* < 0.05).

**Figure 2 molecules-27-07037-f002:**
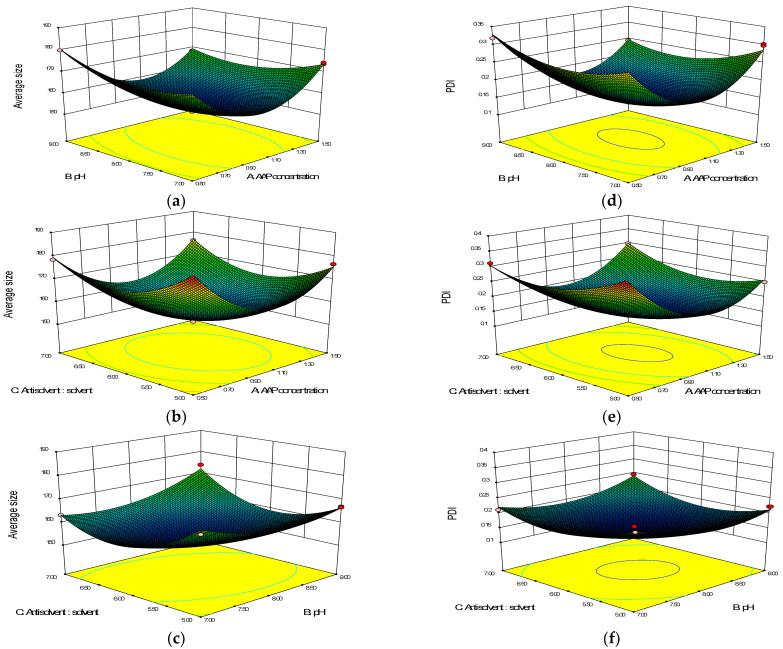
Three-dimensional response surface diagrams of (**a**–**c**) Model 1 and (**d**–**f**) Model 2.

**Figure 3 molecules-27-07037-f003:**
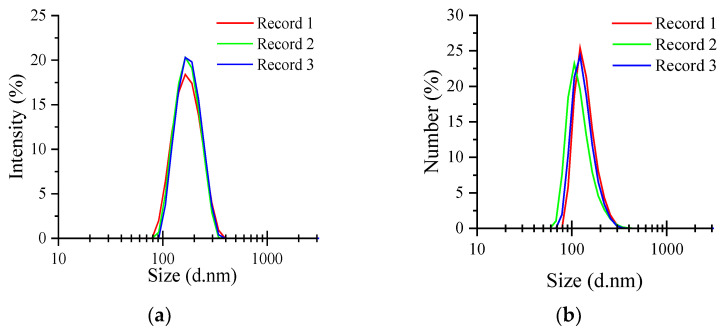
AAP-NP particle size distributions (**a**) intensity; (**b**) number. Records 1 to 3 represent the repeated experiments.

**Figure 4 molecules-27-07037-f004:**
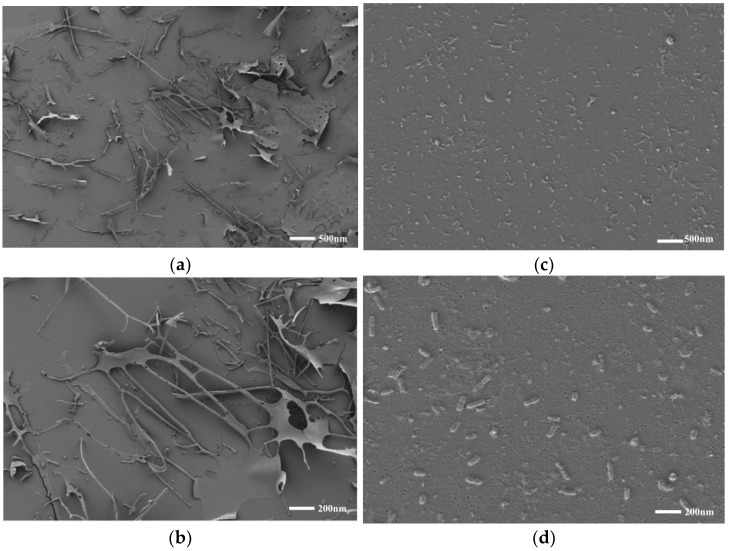
SEM images of AAP (**a**,**b**) and AAP-NPs (**c**,**d**) with scale bars of 500 nm and 200 nm, respectively.

**Figure 5 molecules-27-07037-f005:**
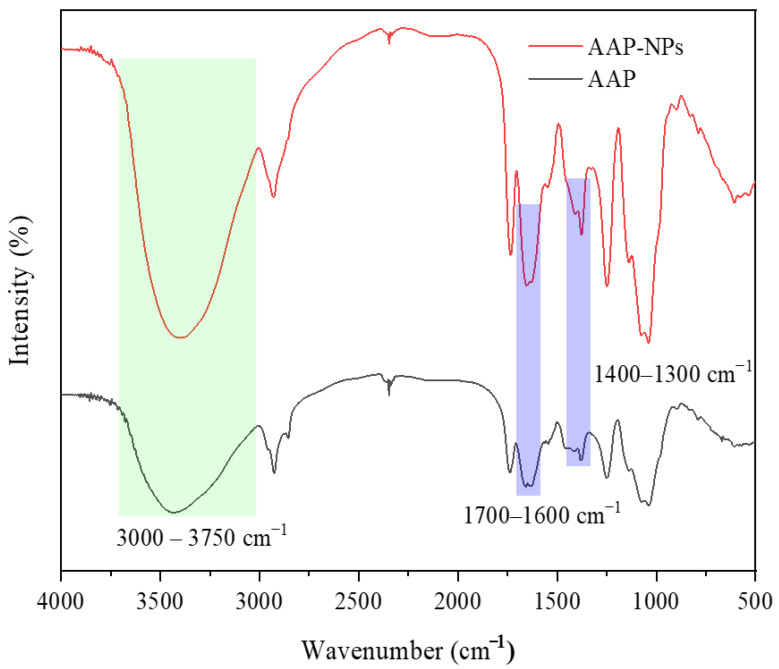
FTIR images of AAP-NPs prepared at the optimal conditions and those of the original AAP.

**Figure 6 molecules-27-07037-f006:**
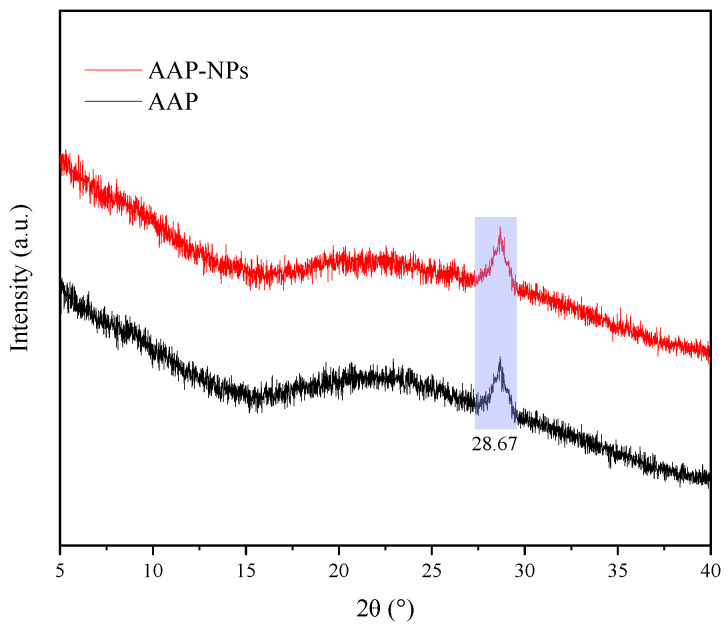
X-ray diffractograms of AAP-NPs and AAP.

**Figure 7 molecules-27-07037-f007:**
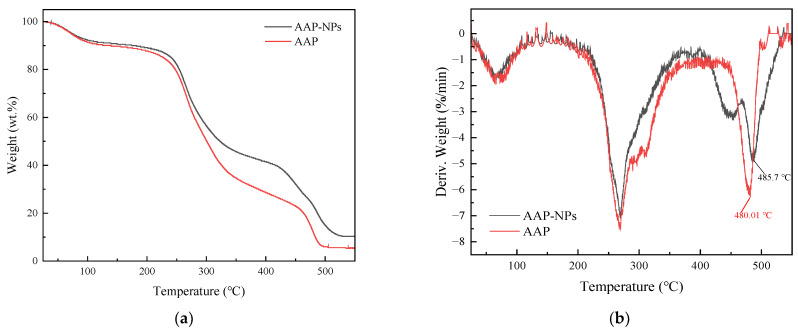
(**a**) Thermogravimetric (TG) and (**b**) derivative thermogravimetric (DTG) curves of AAP-NPs and AAP.

**Figure 8 molecules-27-07037-f008:**
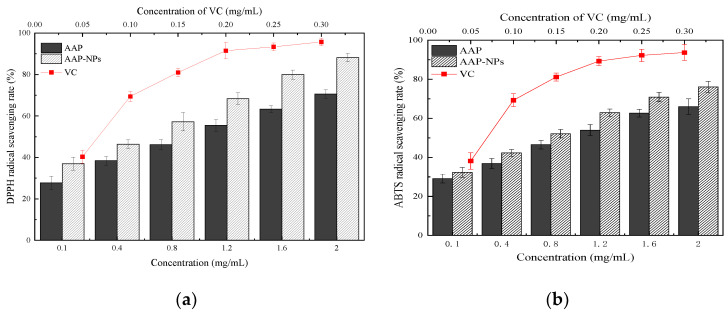
In vitro antioxidant activity of AAP-NPs, original AAP and VC, including (**a**) DPPH radical scavenging activity and (**b**) ABTS+ radical scavenging activity.

**Figure 9 molecules-27-07037-f009:**
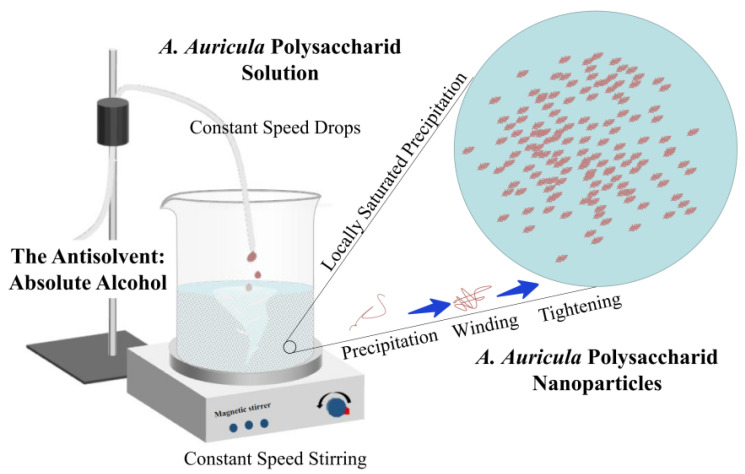
Formation of AAP-NPs via antisolvent precipitation.

**Table 1 molecules-27-07037-t001:** BBD experiment design and results.

No.	Experiment Design			Experimental Results	
	(A) AAP Concentration (%)	(B) pH	(C) Antisolvent: Solvent (*v*:*v*)	(Y_1_) Average Size (nm)	(Y_2_) PDI
1	0.5	8	5:1	184.90	0.352
2	0.5	8	7:1	178.50	0.312
3	0.5	7	6:1	174.30	0.310
4	0.5	9	6:1	179.87	0.318
5	1	7	5:1	169.97	0.247
6	1	9	7:1	173.97	0.245
7	1	9	5:1	166.95	0.223
8	1	7	7:1	163.20	0.212
9	1	8	6:1	153.97	0.142
10	1	8	6:1	152.70	0.136
11	1	8	6:1	153.00	0.138
12	1	8	6:1	154.50	0.142
13	1	8	6:1	151.18	0.155
14	1.5	8	5:1	176.57	0.250
15	1.5	8	7:1	176.27	0.298
16	1.5	7	6:1	173.57	0.302
17	1.5	9	6:1	168.40	0.245

**Table 2 molecules-27-07037-t002:** Variance analysis results of regression models.

Sources of Variance	Model 1 Average Size/(nm)					Model 2 PDI				
	Sum of Squares	*df*	Mean Square	*F*-Value	*p*-Value Prob > *F*	Sum of Squares	*df*	Mean Square	*F*-Value	*p*-Value Prob > *F*
Model	1918.26	9	213.14	99.86	<0.0001 **	0.084	9	9.29 × 10^−3^	58.63	<0.0001 **
A-AAP concentration (%)	49.05	1	49.05	22.98	0.002 **	4.03 × 10^−3^	1	4.03 × 10^−3^	25.46	0.0015 **
B-pH	11.59	1	11.59	5.43	0.0526	3.19 × 10^−3^	1	3.19 × 10^−3^	20.14	0.0028 **
Antisolvent: solvent (*v*:*v*)	12.14	1	12.14	5.69	0.0485 *	4.01 × 10^−5^	1	4.01 × 10^−5^	0.25	0.6303
AB	29.04	1	29.04	13.61	0.0078 **	8.91 × 10^−4^	1	8.91 × 10^−4^	5.63	0.0495 *
AC	9.30	1	9.30	4.36	0.0752	1.94 × 10^−3^	1	1.94 × 10^−3^	12.22	0.01 *
BC	52.09	1	52.09	24.41	0.0017 **	7.17 × 10^−4^	1	7.17 × 10^−4^	4.53	0.071
A^2^	1044.64	1	1044.64	489.43	<0.0001 **	5.20 × 10^−2^	1	5.20 × 10^−2^	328.73	<0.0001 **
B^2^	100.25	1	100.25	46.97	0.0002 **	6.92 × 10^−3^	1	6.92 × 10^−3^	43.69	0.0003 **
C^2^	441.40	1	441.40	206.8	<0.0001 **	1.00 × 10^−2^	1	1.00 × 10^−2^	64.35	<0.0001 **
Residual	14.94	7	2.13			1.11 × 10^−3^	7	1.58 × 10^−4^		
Lack of fit	8.37	3	2.79	1.7	0.3039	8.90 × 10^−4^	3	2.97 × 10^−4^	0.61	0.6434
Pure error	6.57	4	1.64			2.19 × 10^−4^	4	5.48 × 10^−5^		
Total error	1933.20	16				8.08 × 10^−4^	16			
R^2^	0.9923					0.9869				
$R_{adj}^{2}$	0.9823					0.9701				
C.V.%	0.87					5.31				
Adeq. precision	27.915					21.493				

Note: * *p* < 0.05 indicates significant difference; ** *p* < 0.01 indicates a highly significant difference.

**Table 3 molecules-27-07037-t003:** BBD level table and coding of experimental factors.

Factors	Coded Symbols	Levels		
		−1	0	1
AAP concentration	A	0.5	1	1.5
pH value	B	7	8	9
Antisolvent: solvent ratio (*v*/*v*)	C	5:1	6:1	7:1

## Data Availability

Not applicable.

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
