# Peer review of "Formation Optimization, Characterization and Antioxidant Activity of Auricularia auricula-judae Polysaccharide Nanoparticles Obtained via Antisolvent Precipitation"

_molecules, 2022, doi:10.3390/molecules27207037_

Round 1

Reviewer 1 Report

The manuscript describes the synthesis, optimization, characterization, and the antiradical activity of Auricularia auricula-judae polysaccharide nanoparticles obtained by the antisolvent precipitation method. Experimentally, the manuscript has scientific soundness, but there are several issues to consider prior to publication:   

- The introduction section should be completely rewritten to properly highlight the novelty of the work. In the introduction or discussion section, several important articles have been left out (
10.1039/C3TA00050H, 10.1016/j.ijpharm.2016.07.026, 10.1016/j.ajps.2015.10.064, 10.1016/j.foodhyd.2017.07.023), I believe this previous evidence is closely related to this work and worth considering.

- The antioxidant properties involve more than one mechanism of action, the manuscript only evaluates the antiradical activity (radical scavenging) against DPPH and ABTS, change the concept of antioxidant where appropriate.

- Fig 5. “a” and “b” should have the same scale bar, use another suitable image, and replace it. Also, a more complete description of the images in the figure caption might help.

- In pag. 13, the authors pointed out that “Thermostability analysis, thus, lays a foundation for the application of AAP-NPs in high temperature treatments”, due to shift in peak (480.01 to 485.7 °C). Please provide examples.

- Section 3 should be Results and discussion, and section 4. conclusions. Fix.

- Pag. 15 the authors concluded that “These results, thus, indicate that AAP-NPs are suitable for application as antioxidants…” which could be concluded because it was supported by experimental evidence (even though the evaluation was only radical scavenging). But then the authors noted that “…or nanocarriers in food and health-related industries”, which is not possible to conclude based on the current status of the manuscript, so remove the statement, describe it as a potential application that needs further study, or conduct the appropriate experiments to support that conclusion.

Reviewer 2 Report

Manuscript of Ma and co-workers reports the synthesis by antisolvent precipitation, characterization and antioxidant activity of nanoparticles obtained by polysaccharides as starting material.

This work seems to be interesting for the nanochemists community, however, some additional experiments are required before the publication:

·       1.  some additional information about morphology and composition are required. I strongly suggest TEM and XPS analyses. What kind of nanoparticles have been synthesized? Carbon? A fully characterization of these nanosystem is necessary.

·        2. the values reported in Figure 2 have been calculated by SEM measurements? SEM analysis has a resolution of few nanometers (ca. 10 nm, as far as I know), thus how is possible to obtain similar values?

·        3. a definition of "anti-solvent" can be useful for a reader

·        4. probably, the average particle sizes should show only one decimal, due to the sensibility of the method. in addition, significative values should be revised

·        5. from line 242 to 244 the authors describe a characteristic of polysacarides that can be removed from a scientific article

·        6. chapter 4 should be Conclusions

·        7. english must be revised

·         

Round 2

Reviewer 1 Report

The authors have addressed the suggestions. In my opinion, the manuscript is suitable for publication in Molecules.

Reviewer 2 Report

The author addressed the points raised during the first step. Manuscript can be published now